# VAMPnets for deep learning of molecular kinetics

Andreas Mardt[1], Luca Pasquali[1], Hao Wu[1] & Frank Noé [1]

There is an increasing demand for computing the relevant structures, equilibria, and long-timescale kinetics of biomolecular processes, such as protein-drug binding, from high-throughput molecular dynamics simulations. Current methods employ transformation of simulated coordinates into structural features, dimension reduction, clustering the dimension-reduced data, and estimation of a Markov state model or related model of the interconversion rates between molecular structures. This handcrafted approach demands a substantial amount of modeling expertise, as poor decisions at any step will lead to large modeling errors. Here we employ the variational approach for Markov processes (VAMP) to develop a deep learning framework for molecular kinetics using neural networks, dubbed VAMPnets. A VAMPnet encodes the entire mapping from molecular coordinates to Markov states, thus combining the whole data processing pipeline in a single end-to-end framework. Our method performs equally or better than state-of-the-art Markov modeling methods and provides easily interpretable few-state kinetic models.

[1] Department of Mathematics and Computer Science, Freie Universität Berlin, Arnimallee 6, 14195 Berlin, Germany. Andreas Mardt and Luca Pasquali contributed equally to this work. Correspondence and requests for materials should be addressed to F.N. (email: frank.noe@ fu-berlin.de)

The rapid advances in computing power and simulation technologies for molecular dynamics (MD) of biomolecules and fluids[1–4], and ab initio MD of small molecules and materials[5,6], allow the generation of extensive simulation data of complex molecular systems. Thus, it is of high interest to automatically extract statistically relevant information, including stationary, kinetic, and mechanistic properties.

The Markov modeling approach[7–12] has been a driving force in the development of kinetic modeling techniques from MD mass data, chiefly as it facilitates a divide-and-conquer approach to integrate short, distributed MD simulations into a model of the long-timescale behavior. State-of-the-art analysis approaches and software packages[4,13,14] operate by a sequence, or pipeline, of multiple processing steps, that has been engineered by practitioners over the last decade. The first step of a typical processing pipeline is featurization, where the MD coordinates are either aligned (removing translation and rotation of the molecule of interest) or transformed into internal coordinates such as residue distances, contact maps, or torsion angles[4,13,15,16]. This is followed by a dimension reduction, in which the dimension is reduced to much fewer (typically 2–100) slow collective variables, often based on the variational approach or conformation dynamics[17,18], time-lagged independent component analysis (TICA)[19,20], blind source separation[21–23], or dynamic mode decomposition[24–28]—see refs [29,30] for an overview. The resulting coordinates may be scaled, in order to embed them in a metric space whose distances correspond to some form of dynamical distance[31,32]. The resulting metric space is discretized by clustering the projected data using hard or fuzzy data-based clustering methods[11,13,33–36,37], typically resulting in 100–1000 discrete states. A transition matrix or rate matrix describing the transition probabilities or rate between the discrete states at some lag time $\tau$ is then estimated[8,12,38,39] (alternatively, a Koopman model can be built after the dimension reduction[27,28]). The final step toward an easily interpretable kinetic model is coarse-graining of the estimated Markov state model (MSM) down to a few states[40–46].

This sequence of analysis steps has been developed by combining physico-chemical intuition and technical experience gathered in the last ~10 years. Although each of the steps in the above pipeline appears meaningful, there is no fundamental reason why this or any other given analysis pipeline should be optimal. More dramatically, the success of kinetic modeling currently relies on substantial technical expertise of the modeler, as suboptimal decisions in each step may deteriorate the result. As an example, failure to select suitable features in step 1 will almost certainly lead to large modeling errors.

An important step toward selecting optimal models (parameters) and modeling procedures (hyper-parameters) has been the development of the variational approach for conformation dynamics (VAC)[17,18], which offers a way to define scores that measure the optimality of a given kinetic model compared to the (unknown) MD operator that governs the true kinetics underlying the data. The VAC has recently been generalized to the variational approach for Markov processes (VAMP), which allows to optimize models of arbitrary Markov processes, including nonreversible and non-stationary dynamics[47]. The VAC has been employed using cross-validation in order to make optimal hyper-parameter choices within the analysis pipeline described above while avoiding overfitting[34,48]. However, a variational score is not only useful to optimize the steps of a given analysis pipeline, but in fact allows us to replace the entire pipeline with a more general learning structure.

Here we develop a deep learning structure that is in principle able to replace the entire analysis pipeline above. Deep learning has been very successful in a broad range of data analysis and learning problems[49–51]. A feedforward deep neural network is a structure that can learn a complex, nonlinear function $\mathbf{y} = F(\mathbf{x})$. In order to train the network, a scoring or loss function is needed that is maximized or minimized, respectively. Here we develop VAMPnets, a neural network architecture that can be trained by maximizing a VAMP variational score. VAMPnets contain two network lobes that transform the molecular configurations found at a time delay $\tau$ along the simulation trajectories. Compared to previous attempts to include "depth" or "hierarchy" into the analysis method[52,53], VAMPnets combine the tasks of featurization, dimension reduction, discretization, and coarse-grained kinetic modeling into a single end-to-end learning framework. We demonstrate the performance of our networks using a variety of stochastic models and data sets, including a protein-folding data set. The results are competitive with and sometimes surpass the state-of-the-art handcrafted analysis pipeline. Given the rapid improvements of training efficiency and accuracy of deep neural networks seen in a broad range of disciplines, it is likely that follow-up works can lead to superior kinetic models.

## Results

**Variational principle for Markov processes.** Molecular dynamics can be theoretically described as a Markov process $\{\mathbf{x}_t\}$ in the full state space $\Omega$. For a given potential energy function, the simulation setup (e.g., periodic boundaries) and the time-step integrator used, the dynamics are fully characterized by a transition density $p_\tau(\mathbf{x}, \mathbf{y})$, i.e., the probability density that a MD trajectory will be found at configuration $\mathbf{y}$ given that it was at configuration $\mathbf{x}$ a time lag $\tau$ before. Markovianity implies that the $\mathbf{y}$ can be sampled by knowing $\mathbf{x}$ alone, without the knowledge of previous time steps. While the dynamics might be highly nonlinear in the variables $\mathbf{x}_t$, Koopman theory[24,54] tells us that there is a transformation of the original variables into some features or latent variables that, on average, evolve according to a linear transformation. In mathematical terms, there exist transformations to features or latent variables, $\chi_0(\mathbf{x}) = (\chi_{01}(\mathbf{x}), ..., \chi_{0m}(\mathbf{x}))^\top$ and $\chi_1(\mathbf{x}) = (\chi_{11}(\mathbf{x}), ..., \chi_{1m}(\mathbf{x}))^\top$, such that the dynamics in these variables are approximately governed by the matrix $\mathbf{K}$:

$$\mathbb{E}[\chi_1(\mathbf{x}_{t+\tau})] \approx \mathbf{K}^\top \mathbb{E}[\chi_0(\mathbf{x}_t)]. \qquad (1)$$

This approximation becomes exact in the limit of an infinitely large set of features ($m \to \infty$) $\chi_0$ and $\chi_1$, but for a sufficiently large lag time $\tau$ the approximation can be excellent with low-dimensional feature transformations, as we will demonstrate below. The expectation values $\mathbb{E}$ account for stochasticity in the dynamics, such as in MD, but they can be omitted for deterministic dynamical systems[24,26,27].

To illustrate the meaning of Eq. (1), consider the example of $\{\mathbf{x}_t\}$ being a discrete-state Markov chain. If we choose the feature transformation to be indicator functions ($\chi_{0i} = 1$ when $\mathbf{x}_t = i$ and 0 otherwise, and correspondingly with $\chi_{1i}$ and $\mathbf{x}_{t+\tau}$), their expectation values are equal to the probabilities of the chain to be in any given state, $\mathbf{p}_t$ and $\mathbf{p}_{t+\tau}$, and $\mathbf{K} = \mathbf{P}(\tau)$ is equal to the matrix of transition probabilities, i.e., $\mathbf{p}_{t+\tau} = \mathbf{P}^\top(\tau)\mathbf{p}_t$. Previous papers on MD kinetics have usually employed a propagator or transfer operator formulation instead of (1)[7,8]. However, the above formulation is more powerful as it also applies to nonreversible and non-stationary dynamics, as found for MD of molecules subject to external force, such as voltage, flow, or radiation[55,56].

A central result of the VAMP theory is that the best finite-dimensional linear model, i.e., the best approximation in Eq. (1), is found when the subspaces spanned by $\chi_0$ and $\chi_1$ are identical to those spanned by the top $m$ left and right singular functions, respectively, of the so-called Koopman operator[47].

For an introduction to the Koopman operator, please refer to refs. [24,30,54].

How do we choose $\chi_0$, $\chi_1$, and $\mathbf{K}$ from data? First, suppose we are given some feature transformation $\chi_0$, $\chi_1$ and define the following covariance matrices:

$$\mathbf{C}_{00} = \mathbb{E}_t\left[\chi_0(\mathbf{x}_t)\chi_0(\mathbf{x}_t)^\top\right] \tag{2}$$

$$\mathbf{C}_{01} = \mathbb{E}_t\left[\chi_0(\mathbf{x}_t)\chi_1(\mathbf{x}_{t+\tau})^\top\right] \tag{3}$$

$$\mathbf{C}_{11} = \mathbb{E}_{t+\tau}\left[\chi_1(\mathbf{x}_{t+\tau})\chi_1(\mathbf{x}_{t+\tau})^\top\right], \tag{4}$$

where $\mathbb{E}_t[\cdot]$ and $\mathbb{E}_{t+\tau}[\cdot]$ denote the averages that extend over time points and lagged time points within trajectories, respectively, and across trajectories. Then the optimal $\mathbf{K}$ that minimizes the least square error $\mathbb{E}_t\left[\left\|\chi_1(\mathbf{x}_{t+\tau}) - \mathbf{K}^\top\chi_0(\mathbf{x}_t)\right\|^2\right]$ is refs [27,57,47]:

$$\mathbf{K} = \mathbf{C}_{00}^{-1}\mathbf{C}_{01}. \tag{5}$$

Now the remaining problem is how to find suitable transformations $\chi_0$, $\chi_1$. This problem cannot be solved by minimizing the least square error above, as is illustrated by the following example: suppose we define $\chi_0(\mathbf{x}) = \chi_1(\mathbf{x}) = (1(\mathbf{x}))$, i.e., we just map the state space to the constant 1—in this case the least square error is 0 for $\mathbf{K} = [1]$, but the model is completely uninformative as all dynamical information is lost.

Instead, in order to seek $\chi_0$ and $\chi_1$ based on available simulation data, we employ the VAMP theorem introduced in ref. [47], that can be equivalently formulated as the following subspace version.

**VAMP variational principle**. For any two sets of linearly independent functions $\chi_0(\mathbf{x})$ and $\chi_1(\mathbf{x})$, let us call

$$\hat{R}_2[\chi_0,\chi_1] = \left\|\mathbf{C}_{00}^{-\frac{1}{2}}\mathbf{C}_{01}\mathbf{C}_{11}^{-\frac{1}{2}}\right\|_F^2$$

their VAMP-2 score, where $\mathbf{C}_{00}$, $\mathbf{C}_{01}$, $\mathbf{C}_{11}$ are defined by Eqs. (2)–(4) and $\|\mathbf{A}\|_F^2 = n^{-1}\sum_{i,j}A_{ij}^2$ is the Frobenius norm of $n \times n$ matrix $\mathbf{A}$. The maximum value of VAMP-2 score is achieved when the top $m$ left and right Koopman singular functions belong to span $(\chi_{01}, ..., \chi_{0m})$ and span$(\chi_{11}, ..., \chi_{1m})$, respectively.

This variational theorem shows that the VAMP-2 score measures the consistency between subspaces of basis functions and those of dominant singular functions, and we can therefore optimize $\chi_0$ and $\chi_1$ via maximizing the VAMP-2 score. In the special case where the dynamics are reversible with respect to equilibrium distribution the theorem above specializes to variational principle for reversible Markov processes[17,18].

**Learning the feature transformation using VAMPnets**. Here we employ neural networks to find an optimal set of basis functions, $\chi_0(\mathbf{x})$ and $\chi_1(\mathbf{x})$. Neural networks with at least one hidden layer are universal function approximators[58], and deep networks can express strongly nonlinear functions with a fairly few neurons per layer[59]. Our networks use VAMP as a guiding principle and are hence called VAMPnets. VAMPnets consist of two parallel lobes, each receiving the coordinates of time-lagged MD configurations $\mathbf{x}_t$ and $\mathbf{x}_{t+\tau}$ as input (Fig. 1). The lobes have $m$ output nodes and are trained to learn the transformations $\chi_0(\mathbf{x}_t)$ and $\chi_1(\mathbf{x}_{t+\tau})$, respectively. For a given set of transformations, $\chi_0$ and $\chi_1$, we pass a batch of training data through the network and compute the training VAMP score of our choice. VAMPnets bear similarities with auto-encoders[60,61] using a time-delay embedding and are closely related to deep canonical covariance analysis (CCA)[62].

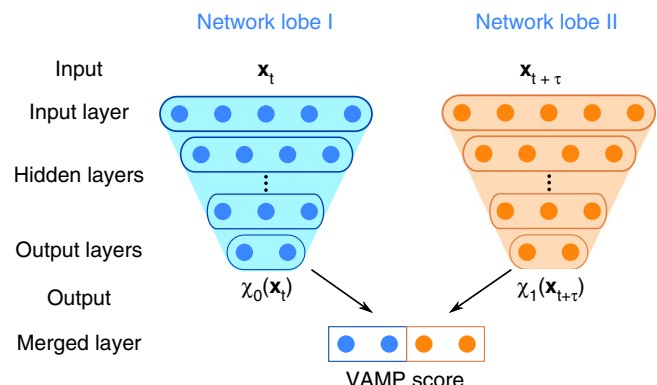

**Fig. 1** Scheme of the neural network architecture used. For each time step $t$ of the simulation trajectory, the coordinates $\mathbf{x}_t$ and $\mathbf{x}_{t+\tau}$ are inputs to two deep networks that conduct a nonlinear dimension reduction. In the present implementation, the output layer consists of a Softmax classifier. The outputs are then merged to compute the variational score that is maximized to optimize the networks. In all present applications, the two network lobes are identical clones, but they can also be trained independently

VAMPnets are identical to deep CCA with time-delay embedding when using the VAMP-1 score discussed in ref. [47], however the VAMP-2 score has easier-to-handle gradients and is more suitable for time series data, due to its direct relation to the Koopman approximation error[47].

The first left and right singular functions of the Koopman operator are always equal to the constant function $1(\mathbf{x}) \equiv 1$[47]. We can thus add 1 to basis functions and train the network by maximizing

$$\hat{R}_2\left[\begin{pmatrix}1\\\chi_0\end{pmatrix}, \begin{pmatrix}1\\\chi_1\end{pmatrix}\right] = \left\|\overline{\mathbf{C}}_{00}^{-\frac{1}{2}}\overline{\mathbf{C}}_{01}\overline{\mathbf{C}}_{11}^{-\frac{1}{2}}\right\|_F^2 + 1, \tag{6}$$

where $\overline{\mathbf{C}}_{00}, \overline{\mathbf{C}}_{01}, \overline{\mathbf{C}}_{11}$ are mean-free covariances of the feature-transformed coordinates:

$$\overline{\mathbf{C}}_{00} = (T-1)^{-1}\overline{\mathbf{X}}\,\overline{\mathbf{X}}^\top \tag{7}$$

$$\overline{\mathbf{C}}_{01} = (T-1)^{-1}\overline{\mathbf{X}}\,\overline{\mathbf{Y}}^\top \tag{8}$$

$$\overline{\mathbf{C}}_{11} = (T-1)^{-1}\overline{\mathbf{Y}}\,\overline{\mathbf{Y}}^\top. \tag{9}$$

Here we have defined the matrices $\mathbf{X} = [X_{ij}] = \chi_{0i}(\mathbf{x}_j) \in \mathbb{R}^{m \times T}$ and $\mathbf{Y} = [Y_{ij}] = \chi_{1i}(\mathbf{x}_{j+\tau}) \in \mathbb{R}^{m \times T}$ with $\{(\mathbf{x}_j, \mathbf{x}_{j+\tau})\}_{j=1}^T$ representing all available transition pairs, and their mean-free versions $\overline{\mathbf{X}} = \mathbf{X} - T^{-1}\mathbf{X}\mathbf{1}$, $\overline{\mathbf{Y}} = \mathbf{Y} - T^{-1}\mathbf{Y}\mathbf{1}$. The gradients of $\hat{R}_2$ are given by:

$$\nabla_{\mathbf{X}}\hat{R}_2 = \frac{2}{T-1}\overline{\mathbf{C}}_{00}^{-1}\overline{\mathbf{C}}_{01}\overline{\mathbf{C}}_{11}^{-1}\left(\overline{\mathbf{Y}} - \overline{\mathbf{C}}_{01}^\top\overline{\mathbf{C}}_{00}^{-1}\overline{\mathbf{X}}\right) \tag{10}$$

$$\nabla_{\mathbf{Y}}\hat{R}_2 = \frac{2}{T-1}\overline{\mathbf{C}}_{11}^{-1}\overline{\mathbf{C}}_{01}^\top\overline{\mathbf{C}}_{00}^{-1}\left(\overline{\mathbf{X}} - \overline{\mathbf{C}}_{01}\overline{\mathbf{C}}_{11}^{-1}\overline{\mathbf{Y}}\right) \tag{11}$$

and are back-propagated to train the two network lobes. See Supplementary Note 1 for derivations of Eqs. (6), (10), and (11).

For simplicity of interpretation, we may just use a unique basis set $\chi = \chi_0 = \chi_1$. Even when using two different basis sets would be meaningful, we can unify them by simply defining $\chi = (\chi_0, \chi_1)^\top$. In this case, we clone the lobes of the network and train them using the total gradient $\nabla\hat{R}_2 = \nabla_{\mathbf{X}}\hat{R}_2 + \nabla_{\mathbf{Y}}\hat{R}_2$.

After training, we asses the quality of the learned features and select hyper-parameters (e.g., network size) while avoiding overfitting using the VAMP-2 validation score

$$\hat{R}_2^{\text{val}} = \left\| \left(\overline{\mathbf{C}}_{00}^{\text{val}}\right)^{-\frac{1}{2}} \overline{\mathbf{C}}_{01}^{\text{val}} \left(\overline{\mathbf{C}}_{11}^{\text{val}}\right)^{-\frac{1}{2}} \right\|_F^2 + 1, \quad (12)$$

where $\overline{\mathbf{C}}_{00}^{\text{val}}, \overline{\mathbf{C}}_{01}^{\text{val}}, \overline{\mathbf{C}}_{11}^{\text{val}}$ are mean-free covariance matrices computed from a validation data set not used during the training.

**Dynamical model and validation.** The direct estimate of the time-lagged covariance matrix $\mathbf{C}_{01}$ is generally nonsymmetric. Hence the Koopman model or MSM $\mathbf{K}$ given by Eq. (5) is typically not time-reversible[28]. In MD, it is often desirable to obtain a time-reversible kinetic model—see[39] for a detailed discussion. To enforce reversibility, $\mathbf{K}$ can be reweighted as described in[28] and implemented in PyEMMA[13]. The present results do not depend on enforcing reversibility, as classical analyses such as PCCA+[63] are avoided as the VAMPnet structure automatically performs coarse graining.

Since $\mathbf{K}$ is a Markovian model, it is expected to fulfill the Chapman–Kolmogorov (CK) equation:

$$\mathbf{K}(n\tau) = \mathbf{K}^n(\tau), \quad (13)$$

for any value of $n \geq 1$, where $\mathbf{K}(\tau)$ and $\mathbf{K}(n\tau)$ indicate the models estimated at a lag time of $\tau$ and $n\tau$, respectively. However, since any Markovian model of MD can be only approximate[8,64], Eq. (13) can only be fulfilled approximately, and the relevant test is whether it holds within statistical uncertainty. We construct two tests based on Eq. (13): in order to select a suitable dynamical model, we proceed as for Markov state models by conducting an eigenvalue decomposition for every estimated Koopman matrix, $\mathbf{K}(\tau)\mathbf{r}_i = \mathbf{r}_i \lambda_i(\tau)$, and computing the implied timescales[9] as a function of lag time:

$$t_i(\tau) = -\frac{\tau}{\ln|\lambda_i(\tau)|}, \quad (14)$$

We chose a value $\tau$, where $t_i(\tau)$ are approximately constant in $\tau$. After having chosen $\tau$, we test whether Eq. (13) holds within statistical uncertainty[65]. For both the implied timescales and the CK test, we proceed as follows: train the neural network at a fixed lag time $\tau^*$, thus obtaining the network transformation $\chi$, and then compute Eq. (13) or Eq. (14) for different values of $\tau$ with a fixed transformation $\chi$. Finally, the approximation of the $i$th eigenfunction is given by

$$\hat{\psi}_i^e(\mathbf{x}) = \sum_j r_{ij} \chi_j(\mathbf{x}). \quad (15)$$

If dynamics are reversible, the singular value decomposition and eigenvalue decomposition are identical, i.e., $\sigma_i = \lambda_i$ and $\psi_i = \psi_i^e$.

**Network architecture and training.** We use VAMPnets to learn molecular kinetics from simulation data of a range of model systems. While any neural network architecture can be employed inside the VAMPnet lobes, we chose the following setup for our applications: the two network lobes are identical clones, i.e., $\chi_0 \equiv \chi_1$, and consist of fully connected networks. In most cases, the networks have less output than input nodes, i.e., the network conducts a dimension reduction. In order to divide the work equally between network layers, we reduce the number of nodes from each layer to the next by a constant factor. Thus, the network architecture is defined by two parameters: the depth $d$ and the number of output nodes $n_{\text{out}}$. All hidden layers employ rectified linear units (ReLU)[66,67].

Here, we build the output layer with Softmax output nodes, i.e., $\chi_i(\mathbf{x}) \geq 0$ for all $i$ and $\sum_i \chi_i(\mathbf{x}) = 1$. Therefore, the activation of an output node can be interpreted as a probability to be in state $i$. As a result, the network effectively performs featurization, dimension reduction, and finally a fuzzy clustering to metastable states, and the $\mathbf{K}(\tau)$ matrix computed from the network-transformed data is the transition matrix of a fuzzy MSM[36,37]. Consequently, Eq. (1) propagates probability distributions in time.

The networks were trained with pairs of MD configurations ($\mathbf{x}_t$, $\mathbf{x}_{t+\tau}$) using the Adam stochastic gradient descent method[68]. For each result, we repeated 100 training runs, each of which with a randomly chosen 90%/10% division of the data into training and validation data. See Methods section for details on network architecture, training, and choice of hyper-parameters.

**Asymmetric double-well potential.** We first model the kinetics of a bistable one-dimensional process, simulated by Brownian dynamics (Methods) in an asymmetric double-well potential (Fig. 2a). A trajectory of 50,000 time steps is generated. Three-layer VAMPnets are set up with 1-5-10-5 nodes in each lobe. The single input node of each lobe is given the current and time-lagged mean-free $x$ coordinate of the system, i.e., $x_t - \mu_1$ and $x_{t+\tau} - \mu_2$, where $\mu_1$ and $\mu_2$ are the respective means, and $\tau = 1$ is used. The network maps to five Softmax output nodes that we will refer to as states, as the network performs a fuzzy discretization by mapping the input configurations to the output activations. The network is trained by using the VAMP-2 score with the four largest singular values.

The network learns to place the output states in a way to resolve the transition region best (Fig. 2b), which is known to be important for the accuracy of a Markov state model[8,64]. This placement minimizes the Koopman approximation error, as seen by comparing the dominant Koopman eigenfunction (Eq. (15)) with a direct numerical approximation of the true eigenfunction obtained by a transition matrix computed for a direct uniform 200-state discretization of the $x$ axis—see ref. [8] for details. The implied timescale and CK tests (Eqs. (13) and (14)) confirm that the kinetic model learned by the VAMPnet successfully predicts the long-time kinetics (Fig. 2c, d).

**Protein-folding model.** While the first example was one-dimensional, we now test if VAMPnets are able to learn reaction coordinates that are nonlinear functions of a multi-dimensional configuration space. For this, we simulate a 100,000 time step Brownian dynamics trajectory (Eq. (17)) using the simple protein-folding model defined by the potential energy function (Supplementary Fig. 1a):

$$U(r) = \begin{pmatrix} -2.5(r-3)^2 & r < 3 \\ 0.5(r-3)^3 - (r-3)^2 & r \geq 3 \end{pmatrix}$$

The system has a five-dimensional configuration space, $\mathbf{x} \in \mathbb{R}^5$, however the energy only depends on the norm of the vector $r = |\mathbf{x}|$. While small values of $r$ are energetically favorable, large values of $r$ are entropically favorable as the number of configurations available on a five-dimensional hypersphere grows dramatically with $r$. Thus, the dynamics are bistable along the reaction coordinate $r$. Four-layer network lobes with 5-32-16-8-2 nodes each were employed and trained to maximize the VAMP-2 score involving the largest nontrivial singular value.

The two output nodes successfully identify the folded and the unfolded states, and use intermediate memberships for the

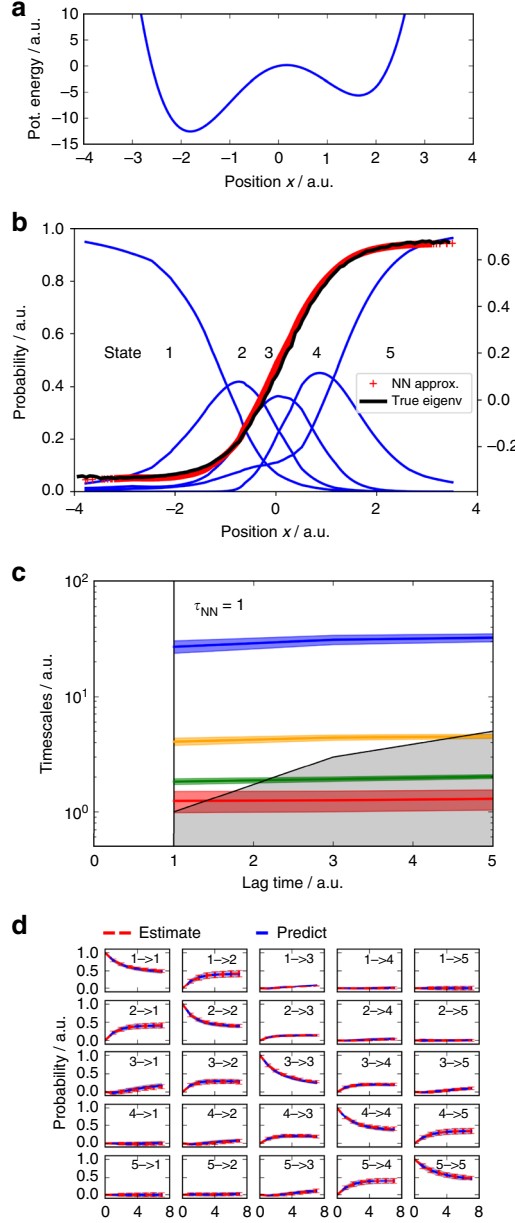

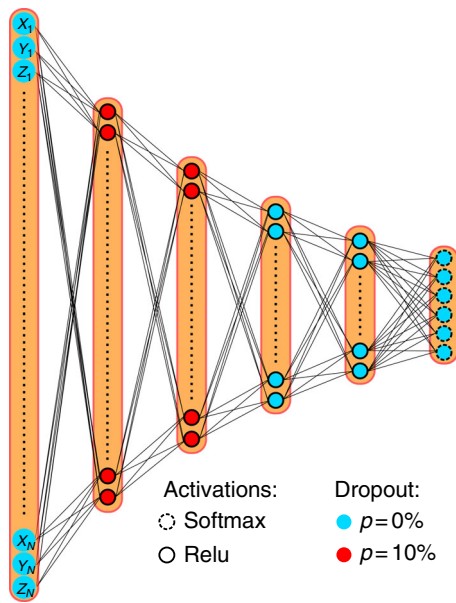

**Fig. 3** Representative structure of one lobe of the VAMPnet used for alanine dipeptide. Here, the five-layer network with six output states used for the results shown in Fig. 4 is shown. Layers are fully connected, have 30-22-16-12-9-6 nodes, and use dropout in the first two hidden layers. All hidden neurons use ReLu activation functions, while the output layer uses Softmax activation function in order to achieve a fuzzy discretization of state space

**Fig. 2** Approximation of the slow transition in a bistable potential. **a** Potential energy function $U(x) = x^4 - 6x^2 + 2x$. **b** Eigenvector of the slowest process calculated by direct numerical approximation (black) and approximated by a VAMPnet with five output nodes (red). Activation of the five Softmax output nodes define the state membership probabilities (blue). **c** Relaxation timescales computed from the Koopman model using the VAMPnet transformation. **d** Chapman–Kolmogorov test comparing long-time predictions of the Koopman model estimated at $\tau = 1$ and estimates at longer lag times. **c**, **d** report 95% confidence interval error bars over 100 training runs

intersecting transition region (Supplementary Fig. 1b). The network excellently approximates the Koopman eigenfunction of the folding process, as apparent from the comparison of the values of the network eigenfunction computed by Eq. (15) with the eigenvector computed from a high-resolution MSM built on the $r$ coordinate (Supplementary Fig. 1b). This demonstrates that the network can learn the nonlinear reaction coordinate mapping $r = |\mathbf{x}|$ based only on maximizing the variational score (Eq. 6). Furthermore, the implied timescales and the CK test indicate that

the network model predicts the long-time kinetics almost perfectly (Supplementary Fig. 1c, d).

**Alanine dipeptide**. As a next level, VAMPnets are used to learn the kinetics of alanine dipeptide from simulation data. It is known that the $\phi$ and $\psi$ backbone torsion angles are the most important reaction coordinates that separate the metastable states of alanine dipeptide, however, our networks only receive Cartesian coordinates as an input, and are thus forced to learn both the nonlinear transformation to the torsion angle space and an optimal cluster discretization within this space, in order to obtain an accurate kinetic model.

A 250 ns MD trajectory generated in ref.[69] (MD setup described there) serves as a data set. The solute coordinates were stored every ps, resulting in 250,000 configurations that are all aligned on the first frame using minimal root mean square deviation fit to remove global translation and rotation. Each network lobe uses the three-dimensional coordinates of the 10 heavy atoms as input, $(x_1, y_1, z_1, ..., x_{10}, y_{10}, z_{10})$, and the network is trained using time lag $\tau = 40$ ps. Different numbers of output states and layer depths are considered, employing the layer sizing scheme described in the Methods section (see Fig. 3 for an example).

A VAMPnet with six output states learns a discretization in six metastable sets corresponding to the free energy minima of the $\phi$/$\psi$ space (Fig. 4b). The implied timescales indicate that given the coordinate transformation found by the network, the two slowest timescales are converged at lag time $\tau = 50$ ps or larger (Fig. 4c). Thus, we estimated a Koopman model at $\tau = 50$ ps, whose Markov transition probability matrix is depicted in Fig. 4d. Note that transition probabilities between state pairs 1 ↔ 4 and 2 ↔ 3 are important for the correct kinetics at $\tau = 50$ ps, but the actual trajectories typically pass via the directly adjacent intermediate states. The model performs excellently in the CK test (Fig. 4e).

**Choice of lag time, network depth, and number of output states.** We studied the success probability of optimizing a VAMPnet with six output states as a function of the lag time $\tau$ by conducting 200 optimization runs. Success was defined as resolving the three slowest processes by finding three slowest timescale higher than 0.2, 0.4, and 1 ns, respectively. Note that the results

shown in Fig. 4 are reported for successful runs in this definition. There is a range of $\tau$ values from 4 to 32 ps where the training succeeds with a significant probability (Supplementary Fig. 2a). However, even in this range the success rate is still below 40%, which is mainly due to the fact that many runs fail to find the rarely occurring third-slowest process that corresponds to the $\psi$ transition of the positive $\phi$ range (Fig. 4b, states 5 and 6).

The breakdown of optimization success for small and large lag times can be most easily explained by the eigenvalue decomposition of Markov propagators[8]. When the lag time exceeds the timescale of a process, the amplitude of this process becomes negligible, making it hard to fit given noisy data. At short lag times, many processes have large eigenvalues, which increases the search space of the neural network and appears to increase the probability of getting stuck in suboptimal maxima of the training score.

We have also studied the success probability, as defined above, as a function of network depth. Deeper networks can represent more complex functions. Also, since the networks defined here reduce the input dimension to the output dimension by a constant factor per layer, deeper networks perform a less radical dimension reduction per layer. On the other hand, deeper networks are more difficult to train. As seen in Supplementary Fig. 2b, a high success rate is found for four to seven layers.

Next, we studied the dependency of the network-based discretization as a function of the number of output nodes (Fig. 5a–c). With two output states, the network separates the state space at the slowest transition between negative and positive values of the $\phi$ angle (Fig. 5a). The result with three output nodes keeps the same separation and additionally distinguishes between the $\alpha$ and $\beta$ regions of the Ramachandran plot, i.e., small and large values of the $\psi$ angle (Fig. 5b). For a higher number of output states, finer discretizations and smaller interconversion timescales are found, until the network starts discretizing the transition regions, such as the two transition states between the $\alpha$ and $\beta$ regions along the $\psi$ angle (Fig. 5c). We chose the lag time depending on the number of output nodes of the network, using $\tau = 200$ ps for two output nodes, $\tau = 60$ ps for three output nodes, and $\tau = 1$ ps for eight output nodes.

A network output with $k$ Softmax neurons describes a $(k-1)$-dimensional feature space as the Softmax normalization removes one degree of freedom. Thus, to resolve $k-1$ relaxation timescales, at least $k$ output nodes or metastable states are required. However, the network quality can improve when given more degrees of freedom in order to approximate the dominant singular functions accurately. Indeed, the best scores using $k = 4$ singular values (three nontrivial singular values) are achieved when using at least six output states that separate each of the six metastable states in the Ramachandran plane (Fig. 5d, e).

For comparison, we investigated how a standard MSM would perform as a function of the number of states (Fig. 5d). For a fair

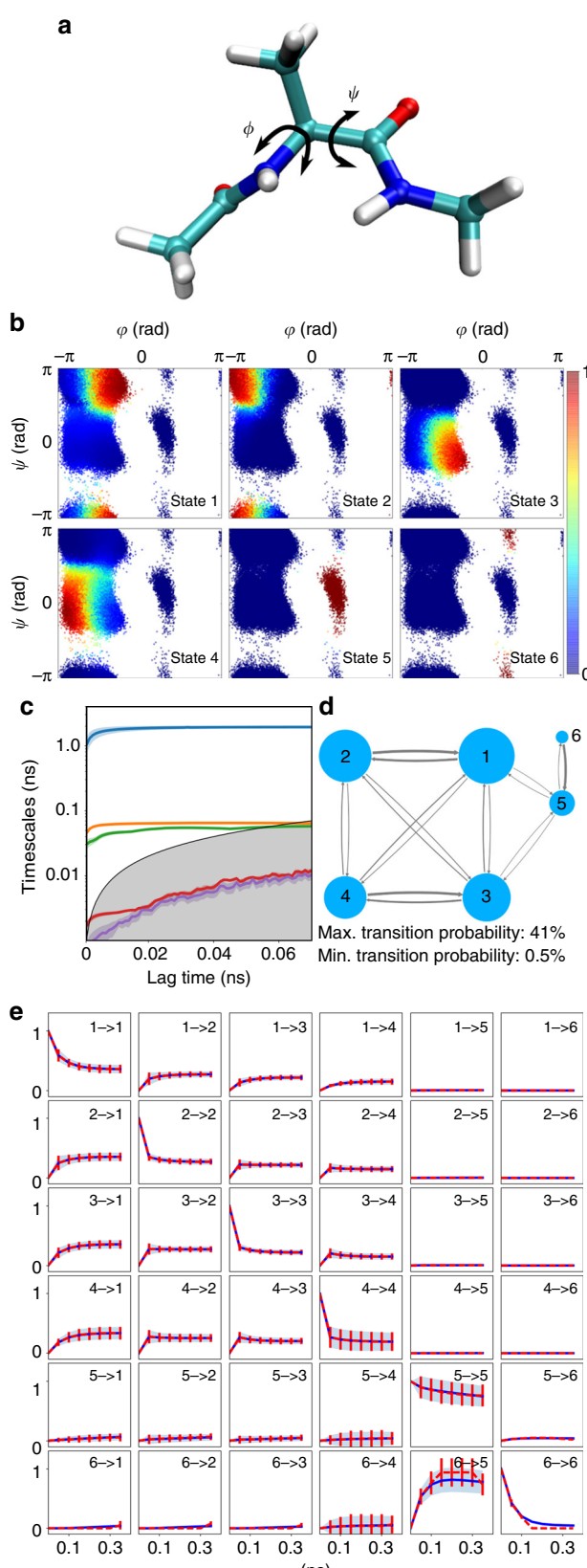

Fig. 4 VAMPnet kinetic model of alanine dipeptide. **a** Structure of alanine dipeptide. The main coordinates describing the slow transitions are the backbone torsion angles $\phi$ and $\psi$, however the neural network inputs are only the Cartesian coordinates of heavy atoms. **b** Assignment of all simulated molecular coordinates, plotted as a function of $\phi$ and $\psi$, to the six Softmax output states. Color corresponds to activation of the respective output neuron, indicating the membership probability to the associated metastable state. **c** Relaxation timescales computed from the Koopman model using the neural network transformation. **d** Representation of the transition probabilities matrix of the Koopman model; transitions with a probability lower than 0.5% have been omitted. **e** Chapman–Kolmogorov test comparing long-time predictions of the Koopman model estimated at $\tau = 50$ ps and estimates at longer lag times. **c, e** report 95% confidence interval error bars over 100 training runs excluding failed runs (see text)

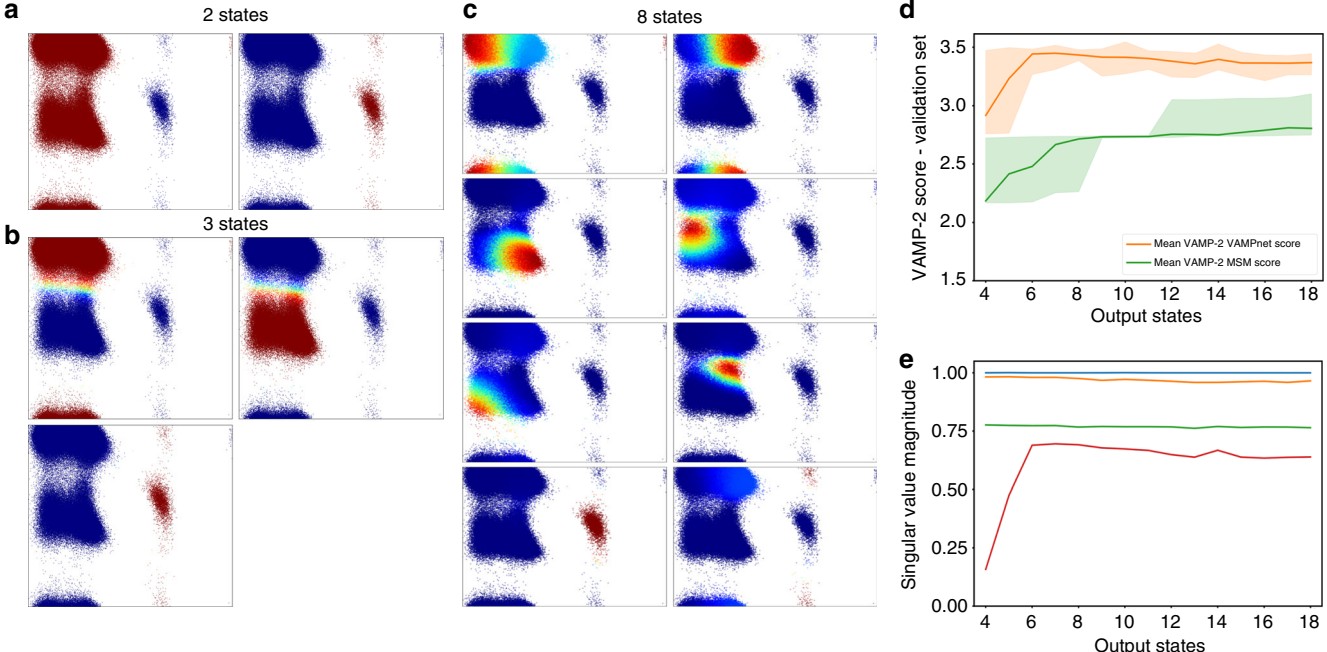

**Fig. 5** Kinetic model of alanine dipeptide as a function of the number of output states. **a–c** Assignment of input coordinates, plotted as a function of $\phi$ and $\psi$, to two, three, and eight output states. Color corresponds to activation of the respective output neuron, indicating the membership probability to this state (Fig. 4b). **d** Comparison of VAMPnet and MSM performance as a function of the number of output states/MSM states. Mean VAMP-2 score and 95% confidence interval from 100 runs are shown. **e** Mean squared values of the four largest singular values that make up the VAMPnets score plotted in **d**

comparison, the MSMs also used Cartesian coordinates as an input, but then employed a state-of-the-art procedure using a kinetic map transformation that preserves 95% of the cumulative kinetic variance[31], followed by $k$-means clustering, where the parameter $k$ is varied. It is seen that the MSM VAMP-2 scores obtained by this procedure is significantly worse than by VAMPnets when <20 states are employed. Clearly, MSMs will succeed when sufficiently many states are used, but in order to obtain an interpretable model, those states must again be coarse-grained onto a fewer-state model, while VAMPnets directly produce an accurate model with few states.

**VAMPnets learn to transform Cartesian to torsion coordinates**. The results above indicate that the VAMPnet has implicitly learned the feature transformation from Cartesian coordinates to backbone torsions. In order to probe this ability more explicitly, we trained a network with 30-10-3-3-2-5 layers, i.e., including a bottleneck of two nodes before the output layer. We find that the activation of the two bottleneck nodes correlates excellently with the $\phi$ and $\psi$ torsion angles that were not presented to the network (Pearson correlation coefficients of 0.95 and 0.92, respectively, Supplementary Fig. 3a, b). To visualize the internal representation that the network learns, we color data samples depending on the free energy minima in the $\phi/\psi$ space they belong to (Supplementary Fig. 3c), and then show where these samples end up in the space of the bottleneck node activations (Supplementary Fig. 3d). It is apparent that the network learns a representation of the Ramachandran plot—the four free energy minima at small $\phi$ values ($\alpha_R$ and $\beta$ areas) are represented as contiguous clusters with the correct connectivity, and are well separated from states with large $\phi$ values ($\alpha_L$ area). The network fails to separate the two substates in the large $\phi$ value range well, which explains the frequent failure to find the corresponding transition process and the third-largest relaxation timescale.

**NTL9 protein-folding dynamics**. In order to proceed to a higher-dimensional problem, we analyze the kinetics of an all-atom protein-folding simulation of the NTL9 protein generated by the Anton supercomputer[1]. A five-layer VAMPnet was trained at lag time $\tau = 10$ ns using 111,000 time steps, uniformly sampled from a 1.11 ms trajectory. Since NTL9 is folding and unfolding, there is no unique reference structure to align Cartesian coordinates to—hence we use internal coordinates as a network input. We computed the nearest-neighbor heavy-atom distance, $d_{ij}$ for all non-redundant pairs of residues $i$ and $j$ and transformed them into contact maps using the definition $c_{ij} = \exp(-d_{ij})$, resulting in 666 input nodes.

Again, the network performs a hierarchical decomposition of the molecular configuration space when increasing the number of output nodes. Figure 6a shows the decomposition of state space for two and five output nodes, and the corresponding mean contact maps and state probabilities. With two output nodes, the network finds the folded and unfolded state that are separated by the slowest transition process (Fig. 6a, middle row). With five output states, the folded state is decomposed into a stable and well-defined fully folded substate and a less stable, more flexible substate that is missing some of the tertiary contacts compared to the fully folded substate. The unfolded substate decomposes into three substates, one of them largely unstructured, a second one with residual structure, thus forming a folding intermediate, and a mis-folded state with an entirely different fold including a non-native $\beta$-sheet.

The relaxation timescales found by a five-state VAMPnet model are en par with those found by a 40-state MSM using state-of-the-art estimation methods (Fig. 6b, c). However, the fact that only five states are required in the VAMPnet model makes it easier to interpret and analyze. Additionally, the CK test indicates excellent agreement between long-time predictions and direct estimates.

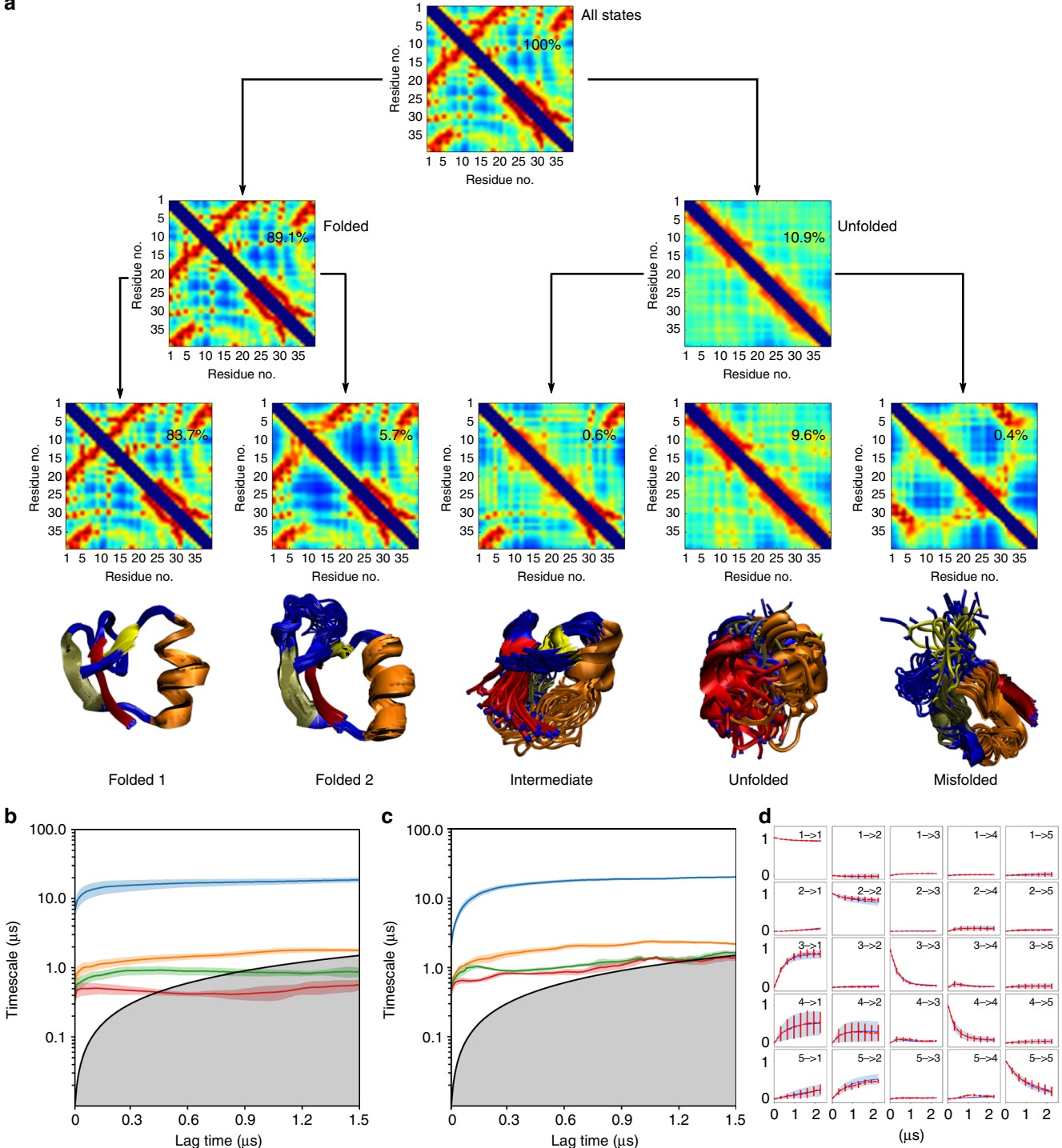

**Fig. 6** VAMPnet results of NTL9-folding kinetics. **a** Hierarchical decomposition of the NTL9 protein state space by a network with two and five output nodes. Mean contact maps are shown for all MD samples grouped by the network, along with the fraction of samples in that group. 3D structures are shown for the five-state decomposition, residues involved in $\alpha$-helices or $\beta$-sheets in the folded state are colored identically across the different states. **b** Relaxation timescales computed from the Koopman model approximated using the transformation applied by a neural network with five output nodes. **c** Relaxation timescales from a Markov state model computed from a TICA transformation of the contact maps, followed by k-means clustering with $k = 40$. **d** Chapman–Kolmogorov test comparing long-time predictions of the Koopman model estimated at $\tau = 320$ ns and estimates at longer lag times. **b**–**d** report 95% confidence interval error bars over 100 training runs

## Discussion

We have introduced a deep learning framework for molecular kinetics, called VAMPnet. Data-driven learning of molecular kinetics is usually done by shallow learning structures, such as TICA and MSMs. However, the processing pipeline, typically consisting of featurization, dimension reduction, MSM estimation, and MSM coarse-graining is, in principle, a handcrafted deep learning structure. Here we propose to replace the entire pipeline by a deep neural network that learns optimal feature transformations, dimension reduction and, if desired, maps the

MD time steps to a fuzzy clustering. The key to optimize the network is the VAMP variational approach that defines scores by which learning structures can be optimized to learn models of both equilibrium and non-equilibrium MD.

Although MSM-based kinetic modeling has been refined over more than a decade, VAMPnets perform competitively or superior in our examples. In particular, they perform extremely well in the Chapman–Kolmogorov test that validates the long-time prediction of the model. VAMPnets have a number of advantages over models based on MSM pipelines: (i) they may be overall more optimal, because featurization, dimension reduction, and clustering are not explicitly separate processing steps. (ii) When using Softmax output nodes, the VAMPnet performs a fuzzy clustering of the MD structures fed into the network and constructs a fuzzy MSM, which is readily interpretable in terms of transition probabilities between metastable states. In contrast to other MSM coarse-graining techniques, it is thus not necessary to accept reduction in model quality in order to obtain a few-state MSM, but such a coarse-grained model is seamlessly learned within the same learning structure. (iii) VAMPnets require less user expertise to train than an MSM-based processing pipelines, and the formulation of the molecular kinetics as a neural network learning problem enables us to exploit an arsenal of highly developed and optimized tools in learning softwares such as tensorflow, theano, or keras.

Despite these advantages, VAMPnets still miss many of the benefits that come with extensions developed for the MSM approach. This includes multi-ensemble Markov models that are superior to single conventional MSMs in terms of sampling rare events by combining data from multiple ensembles[70–75], augmented Markov models that combine simulation data with experimental observation data[76], and statistical error estimators developed for MSMs[77–79]. Since these methods explicitly use the MSM likelihood, it is currently unclear, how they could be implemented in a deep learning structure such as a VAMPnet. Extending VAMPnets toward these special capabilities is a challenge for future studies.

Finally, a remaining concern is that the optimization of VAMPnets can get stuck in suboptimal local maxima. In other applications of network-based learning, a working knowledge has been established to find which type of network implementation and learning algorithm are most suitable for robust and reproducible learning. For example, it is conceivable that the VAMPnet lobes may benefit from convolutional filters[80] or different types of transfer functions. Suitably chosen convolutions, as in ref. [81] may also lead to learned feature transformations that are transferable within a given class of molecules.

## Methods

**Neural network structure**. Each network lobe in Fig. 1 has a number of input nodes given by the data dimension. According to the VAMP variational principle (Sec. A), the output dimension must be at least equal to the number of Koopman singular functions that we want to approximate, i.e., equal to $k$ used in the score function $\hat{R}_2$. In most applications, the number of input nodes exceeds the number of output nodes, i.e., the network conducts a dimension reduction. Here, we keep the dimension reduction from layer $i$ with $n_i$ nodes to layer $i + 1$ with $n_{i+1}$ nodes constant:

$$\frac{n_i}{n_{i+1}} = \left(\frac{n_{\text{in}}}{n_{\text{out}}}\right)^{1/d}, \tag{16}$$

where $d$ is the network depth, i.e., the number of layers excluding the input layer. Thus, the network structure is fixed by $n_{\text{out}}$ and $d$. We tested different values for $d$ ranging from 2 to 11; for alanine dipeptide, Supplementary Fig. 2b reports the results in terms of the training success rate described in the Results section. Networks have a number of parameters that ranges between 100 and 400,000, most of which are between the first and second layer due to the rapid dimension reduction of the network. To avoid overfitting, we use dropout during training[82], and select hyper-parameters using the VAMP-2 validation score.

**Neural network hyper-parameters**. Hyper-parameters include the regularization factors for the weights of the fully connected and the Softmax layer, the dropout probabilities for each layer, the batch size, and the learning rate for the Adam algorithm. Since a grid search in the joint parameter space would have been too computationally expensive, each hyper-parameter was optimized using the VAMP-2 validation score while keeping the other hyper-parameters constant. We started with the regularization factors due to their large effect on the training performance, and observed optimal performance for a factor of $10^{-7}$ for the fully connected hidden layers and $10^{-8}$ for the output layer; regularization factors $>10^{-4}$ frequently led to training failure. Subsequently, we tested the dropout probabilities with values ranging from 0 to 50% and found 10% dropout in the first two hidden layers and no dropout otherwise to perform well. The results did not strongly depend on the training batch size, however, more training iterations are necessary for large batches, while small batches exhibit stronger fluctuations in the training score. We found a batch size of 4000 to be a good compromise, with tested values ranging between 100 and 16,000. The optimal learning rate strongly depends on the network topology (e.g., the number of hidden layers and the number of output nodes). In order to adapt the learning rate, we started from an arbitrary rate of 0.05. If no improvement on the validation VAMP-2 score was observed over 10 training iterations, the learning rate was reduced by a factor of 10. This scheme led to better convergence of the training and validation scores and better kinetic model validation compared to using a high learning rate throughout.

The time lag between the input pairs of configurations was selected depending on the number of output nodes of the network: larger lag times are better at isolating the slowest processes, and thus are more suitable with a small number of output nodes. The procedure of choosing network structure and lag time is thus as follows: First, the number of output nodes $n$ and the hidden layers are selected, which determines the network structure as described above. Then, a lag time is chosen in which the largest $n$ singular values (corresponding to the $n - 1$ slowest processes) can be trained consistently.

**VAMPnet training and validation**. We pre-trained the network by minimizing the negative VAMP-1 score during the first third of the total number of epochs, and subsequently optimize the network with VAMP-2 optimization (Sec. B). In order to ensure robustness of the results, we performed 100 network optimization runs for each problem. In each run, the data set was shuffled and randomly split into 90%/10% for training and validation, respectively. To exclude outliers, we then discarded the best 5% and the worst 5% of results. Hyper-parameter optimization was done using the validation score averaged over the remaining runs. Figures report training or validation mean and 95% confidence intervals.

**Brownian dynamics simulations**. The asymmetric double well and the protein-folding toy model are simulated by over-damped Langevin dynamics in a potential energy function $U(\mathbf{x})$, also known as Brownian dynamics, using an forward Euler integration scheme. The position $\mathbf{x}_t$ is propagated by time step $\Delta t$ via:

$$\mathbf{x}_{t+\Delta t} = \mathbf{x}_t - \Delta t \frac{\nabla U(\mathbf{x})}{kT} + \sqrt{2\Delta t D}\mathbf{w}_t, \tag{17}$$

where $D$ is the diffusion constant and $kT$ is the Boltzmann constant and temperature. Here, dimensionless units are used and $D = 1$, $kT = 1$. The elements of the random vector $\mathbf{w}_t$ are sampled from a normal distribution with zero mean and unit variance.

**Hardware used and training times**. VAMPnets were trained on a single NVIDIA GeForce GTX 1080 GPU, requiring between 20 seconds (for the double-well problem) and 180 seconds for NTL9 for each run.

**Code availability**. TICA, $k$-means and MSM analyses were conducted with PyEMMA version 2.4, freely available at http://www.pyemma.org. VAMPnets are implemented using the freely available packages keras[83] with tensorflow-gpu[84] as a backend. The code can be obtained at https://github.com/markovmodel/deeptime.

**Data availability**. Data for NTL9 can be requested from the authors of ref. [1]. Data for all other examples is available at https://github.com/markovmodel/deeptime.

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

## Acknowledgements

We are grateful to Cecilia Clementi, Robert T. McGibbon, and Max Welling for valuable discussions. This work was funded by Deutsche Forschungsgemeinschaft (SFB958/A04, Transregio 186/A12, SFB 1114/A4, NO 825/4–1 as part of research group 2518) and European Research Commission (ERC StG 307494 "pcCell").

## Author contributions

A.M. and L.P. conducted research and developed software. H.W. and F.N. designed research and developed theory. All authors wrote the paper.

## Additional information

**Competing interests:** The authors declare no competing financial interests.

