## [Peer Review File · Nature Communications]

Reviewers' comments:

Reviewer #1 (Remarks to the Author):

The authors present a coherent Machine Learning approach to the problem of modeling the kinetics in molecular dynamics simulations. The paper proposes to replace the series of manual steps currently employed in the field (featurization, dimensionality reduction, discretization, and coarse grained kinetic modelling) with a single Deep Learning network. It is an elegant approach to the problem, and convincing analyses and results are presented on a range of systems. In addition to the main results in the paper, the authors put effort into exploring various aspects of the trained networks, for instance the ability of one of the networks to automatically learn the translation from cartesian to internal coordinates, which is interesting in its own right. In my opinion, this work represents a significant advance to the field, and I recommend the paper be accepted conditioned on the minor revisions suggested below.

Page 2, right column, top

It was not quite clear to me what you meant by "infinitely-sized feature transformations". Do you mean an infinite set of features (i.e., $m \rightarrow \infty$?), or are you referring to some property of the transformations themselves. Please clarify.

Page 4, right column

You write that training was done on 90% of the dataset, while 10% was used for validation. Since the validation set is used for tuning hyperparameters, it seems that there is a risk of overfitting. The standard procedure is to allocate a third 'test' set for the final evaluation of the model, to guard against overfitting on the validation set. Could you clarify why you didn't use this approach here?

Page 4, right column, Result section A

In the introduction, VAMPnets are proposed as a technique that could potentially replace manual MSM construction, which is described as requiring 'substantial technical expertise'. It is therefore surprising that you in the analysis of this first test case use an MSM as your ground truth (Figure 2b states "True Eigenv" in the caption). Could you provide the reader a reason to trust in this "high-resolution MSM - either by providing a reference (if it has been published previously), or a description of how it was estimated and validated?

Regarding the results of this first test case, the approximation of the second eigenvalue seems a little disappointing (fig 2b). It is also substantially worse than in the second case (Suppl fig 1b). Could you comment on whether this is simply due to the fact that you used a smaller network?

Page 6, left column

Story-wise, the transition from the results in Fig 4, and the subsequent "Dependency of training efficiency on the lag time" is a bit unclear to me. First you report excellent results for an output size of 6, and then you systematically investigate the dependency on τ , but on an output size of 4. In this latter case, you then report that only 11% of optimizations lead to the correct result. Does the same apply for the 6-output case that you described in Figure 4? And if so, does this mean that the results you show in Figure 4 were hand-selected from a larger set of optimizations where 90% failed? It is also not clear to me why you switch between an output size of 6 to and output size of 4. Please clarify.

Page 6, left column

In the text, you write "VAMP-net with four output set", and refer to figure 5c. However 5c reports on models with 8 output nodes. Is this a mistake?
(also, please check the sentence itself - should "set" be "states"?)

Details:

Figure 4 is referred to before 3 in the text.

Reviewer #2 (Remarks to the Author):

The manuscript by Mardt et al. describes a deep learning methodology to replace several steps involved in the preparation of a MSM. The study is thorough, of interest to a wide community, and seems to bear useful results. The deep learning approach, by integrating different steps from feature selection to filtering may well help in streamlining the overall process. While the authors acknowledge limitations in terms of advanced extensions (e.g., reweighting), the absence of a clear likelihood function somehow obscures the procedure. In particular, to what extent is detailed balance enforced by the deep learning algorithm? Though the simple examples may well do without explicit constraint of detailed balance, the last one (NTL9) surely requires it.

Also, is there a way to automatically tell the optimal number of output states that should be selected, or does this remain the choice of the user?

Reviewer #3 (Remarks to the Author):

This paper is a significant advance in the field of long timescale molecular dynamics simulation. The authors trained deep neural networks end-to-end the model the empirical process of constructing MSMs.

Here are my comments, in no particular order of importance.

1) It seems that the network topology (number of layers, number of neurons per layer) changes depending on the problem. The author did not explain how they designed the various network configurations. In other words, how did the authors optimize the network design? In some parts of the paper, it is hinted that domain knowledge was used in network design. Yet at the end, it looks like they did some rudimentary guided search. If it was a guided search, they should also expand on the details (min layer tested, max layers tested, min neurons tested, max neurons tested, dropout values tested, etc...)

2) A more specific example is the peptide neural network that had 30-10-3-3-2-5 layers, including a bottleneck of two nodes before the output layer that correspond to torsion angles. I would like the authors to explore what happens in terms of accuracy when the neural network design changes when it is not guided by expert knowledge (do not introduce a bottleneck layer based on no. of torsions). This extra information will also demonstrate how sensitive the network performance is to minor nuances to network architecture.

3) How do you determine the number of states in the final softmax layer? It seems to be the authors are setting the final layer based on what they are "expecting" to observe. How does the model accuracy fare when that number of final states is set to an "incorrect" number?

4) The network has 2 lobes, are they trained independently from one another (i.e. do the weights

of lobe 1 share the same values as the weights of lobe 2)?

5) Why was dropout only used for the first 2 layers? The usual practice is to apply dropout to all layers.

6) Can the trained neural network for NTL9 protein folding be directly used to generate MSMs of other proteins simulated under the same settings? Specifically, can the NTL9 neural network be used in inferencing mode (without retraining) on the other dozen or so proteins that DE Shaw simulated? If the results are not good, the authors should look into transfer learning using the NTL9 network.

7) The authors mentioned in their abstract that their method "are competitive or outperform state-of-the-art Markov modeling methods". While they have provided convincing qualitative evidence of how VAMPnets can be used in place of MSMs methods, it is unclear how it is outperforming current MSM methods. The authors should perform a more quantitative comparison based on some error metric, or they should remove any claims of their methods performance relative to MSMs.

8) I would like the authors to address the time it takes to train the model (preferably the time including model exploration and network design as well). This should be stated in comparison with the current manual engineering process of creating MSMs.

9) Was there any k-fold cross-validation performed at all? I have the impression the authors did a 90/10 train/validation split only did not do k-fold cross-validation. This is problematic, the authors should be doing at least 5-fold cross validation to ensure their results are robust and not because they were "lucky" in splitting the data "correctly".

10) Also, how was the split between training and validation performed? Did they do a time-index split, or did they randomly shuffle the frames and then do the split?

11) It seems that the authors performed some guided search on the hyper-parameters using the VAMP-2 validation score, however they now risk overfitting the hyper-parameters (aka neural network design) against the validation set, and I presume the validation set is what the authors are using to present all the results in the various figures. The authors should perform additional experiments but using a 3-way train/validation/test split. Keep the same protocol, but evaluate the final results on a separate test set that is not used in both the training of the model, or during hyperparameter optimization.

Reviewer #1

Comments: The authors present a coherent Machine Learning approach to the problem of modeling the kinetics in molecular dynamics simulations. The paper proposes to replace the series of manual steps currently employed in the field (featurization, dimensionality reduction, discretization, and coarse grained kinetic modelling) with a single Deep Learning network. It is an elegant approach to the problem, and convincing analyses and results are presented on a range of systems. In addition to the main results in the paper, the authors put effort into exploring various aspects of the trained networks, for instance the ability of one of the networks to automatically learn the translation from cartesian to internal coordinates, which is interesting in it's own right. In my opinion, this work represents a significant advance to the field, and I recommend the paper be accepted conditioned on the minor revisions suggested below.

We thank the referee for the supporting assessment.

- *It was not quite clear to me what you meant by "infinitely-sized feature transformations". Do you mean an infinite set of features (i.e., $m \rightarrow \infty$?), or are you referring to some property of the transformations themselves. Please clarify.*

What is meant is that the number of features goes to infinity, i.e. $m \rightarrow \infty$. This is clarified in the revised article.

In the special case of a Markov state model, imagine that the number of Markov states go to infinity – in this limit the transition density $p_\tau(\mathbf{x}, \mathbf{y})$ is exactly described. A similar argument holds true for other types of features.

- *You write that training was done on 90% of the dataset, while 10% was used for validation. Since the validation set is used for tuning hyperparameters, it seems that there is a risk of overfitting. The standard procedure is to allocate a third 'test' set for the final evaluation of the model, to guard against overfitting on the validation set. Could you clarify why you didn't use this approach here?*

Apologies for the incomplete explanation: For every choice made and every result reported we performed 100 optimization runs. In each run, the 90% / 10% split of data into training and validation sets was made randomly. The validation score is obtained as an average over these shuffled data splits. Similar as cross-validation, this approach ensures that the results are robust with respect to data splitting. We have described this procedure in the revised Methods section.

- *In the introduction, VAMPnets are proposed as a technique that could potentially replace manual MSM construction, which is described as requiring 'substantial technical expertise'. It is therefore surprising that you in the analysis of this first test case use an MSM as your ground truth (Figure 2b states "True Eigenv" in the caption). Could you provide the reader a reason to trust in this "high-resolution MSM - either by providing a reference (if it has been published previously), or a description of how it was estimated and validated?*

Our caption was indeed confusing. Now we refer to the ground truth as “direct numerical approximation”. Since this example is one-dimensional, it is possible to approximate the true eigenvector to arbitrary accuracy by simply dividing the x -Axis in many small boxes, computing transition probabilities between them, and computing the resulting eigenvector of the slowest process. For a similar problem, this was done e.g. in Prinz et al., J. Chem. Phys. 134:174105 (2011). Technically, this *is* an MSM between the discrete boxes. However, it is quite different from an MSM of a high-dimensional macromolecular dynamics, where this exhaustive discretization cannot be made. In the latter situation, one needs to make many design decisions, such as selecting a subset of coordinates to work with, feature functions, a relatively coarse discretization, a lag time, etc. This latter type of MSM is what we intend to replace by VAMPnets.

Figure 1: Training success probability of VAMPnets for alanine dipeptide with (a) 4 output states and (b) 6 output states.

- Regarding the results of this first test case, the approximation of the second eigenvalue seems a little disappointing (fig 2b). It is also substantially worse than in the second case (Suppl fig 1b). Could you comment on whether this is simply due to the fact that you used a smaller network?

Thank you for prompting us to re-examine this issue. We found that this rather coarse eigenvector approximation is the result of an optimization issue, i.e. the network was stuck in a local minimum. We were able to consistently get results with smooth membership functions and excellent Chapman-Kolmogorow validation by modifying our optimization protocol. Now we pre-train the network using the VAMP-1 loss function, and subsequently optimize it with VAMP-2. Please note that this approach is currently a heuristic, and we do not (yet) have a rigorous explanation why it consistently performs better for the problem at hand. The new procedure is described in the revised Methods section.

- Story-wise, the transition from the results in Fig 4, and the subsequent "Dependency of training efficiency on the lag time" is a bit unclear to me. First you report excellent results for an output size of 6, and then you systematically investigate the dependency on tau, but on an output size of 4. In this latter case, you then report that only 11% of optimizations lead to the correct result. Does the same apply for the 6-output case that you described in Figure 4? And if so, does this mean that the results you show in Figure 4 were hand-selected from a larger set of optimizations where 90% failed? It is also not clear to me why you switch between an output size of 6 to and output size of 4. Please clarify.

The reason why we had shown the τ dependency with only 4 output states is that for this setting the failure rate was especially high. However, we agree that consistently using the same number of states is better for clarity, and thus we have recomputed the τ dependency with 6 output nodes. As shown by subsequent plot, the success rate changes from a maximum of 10% with 4 output states to 40% with 6 output states. We have included the new results as revised Supplementary Figure 2. The results shown in Fig. 4 were obtained from the 40% successful runs out of a total of 200 attempts. This selection was not made manually, but automatically using the criteria of success described in the section "Dependence of training efficiency on the lag time". The difference between failure and success is very clear as in the case of a failure the third timescale is much smaller than 0.2 ns. We added this information to the revised manuscript.

- In the text, you write "VAMP-net with four output set", and refer to figure 5c. However 5c reports on models with 8 output nodes. Is this a mistake? (also, please check the sentence itself - should "set" be "states"?)
- Details: Figure 4 is referred to before 3 in the text.

Thank you. We have corrected these issues in the revised version.

Reviewer #2:

Comments: The manuscript by Mardt et al. describes a deep learning methodology to replace several steps involved in the preparation of a MSM. The study is thorough, of interest to a wide community, and seems to bear useful results. The deep learning approach, by integrating different steps from feature selection to filtering may well help in streamlining the overall process.

We thank the referee for the supporting assessment.

- While the authors acknowledge limitations in terms of advanced extensions (e.g., reweighting), the absence of a clear likelihood function somehow obscures the procedure. In particular, to what extent is detailed balance enforced by the deep learning algorithm? Though the simple examples may well do without explicit constraint of detailed balance, the last one (NTL9) surely requires it.

If fact there is no need for detailed balance in any of the present results and the analyses done here (e.g. calculation of implied timescales, Chapman-Kolmogorow tests). In general, detailed balance is needed for specific analyses, such as the PCCA+ method (Röblitz and Weber, Adv Data Anal. Classif. 7, 147-179, 2013) that is frequently used to coarse-grain MSMs (see discussion in Trendelkamp-Schroer et al, JCP 143: 174101, 2015). However, coarse-graining is done by our VAMPnet architecture automatically. We think it is a benefit of VAMPnets is that the variational optimization does not rely on detailed balance, i.e. VAMPnets can be used to analyze dynamics that are not reversible, and not even stationary. This has become possible with the recently developed VAMP approach.

However, in some cases it is clearly desirable to have a resulting \mathbf{K} matrix that fulfills detailed balance. This may be done at the analysis level. Fortunately, obtaining a reversible \mathbf{K} is possible without a likelihood function: In Wu et al., JCP 146:154104 (2017), a reweighting procedure is described in which an arbitrary Koopman model (and that includes Markov state models) can be made reversible without introducing the estimation bias that would arise from forcefully symmetrizing the estimated \mathbf{C}_{01} matrix. This procedure works by re-weighting the data to the stationary distribution of the estimated Koopman model, and then symmetrizing the time-covariance matrix \mathbf{C}_{01} estimated from the reweighted data.

In the revised manuscript, we have referred to this approach in the section “Dynamical model and validation”. As we do not use detailed balance in this paper, we prefer to stay brief about this technical issue and explore it in more detail elsewhere.

- Also, is there a way to automatically tell the optimal number of output states that should be selected, or does this remain the choice of the user?

As shown in Fig. 5d, the VAMP score depends on the number of output states, but it reaches a plateau when all the states that are metastable at the chosen lag time are resolved. This is an effective automatic selection criterion. However, the user can choose to have a different number of output states in order to enforce a certain level of granularity - see Figures 5a-c and Figure 6a.

Reviewer #3:

Comments: This paper is a significant advance in the field of long timescale molecular dynamics simulation. The authors trained deep neural networks end-to-end the model the empirical process of constructing MSMs.

We thank the referee for the supporting assessment.

- *It seems that the network topology (number of layers, number of neurons per layer) changes depending on the problem. The author did not explain how they designed the various network configurations. In other words, how did the authors optimize the network design? In some parts of the paper, it is hinted that domain knowledge was used in network design. Yet at the end, it looks like they did some rudimentary guided search. If it was a guided search, they should also expand on the details (min layer tested, max layers tested, min neurons tested, max neurons tested, dropout values tested, etc...)*

Indeed we have restricted the search to a certain class of network architectures. For all molecular examples, we have chosen the “cone” shaped design in which we started with a number of input nodes n_{in} defined by the input coordinates, end with a number of output nodes n_{out} , and reduce the number of nodes in each of the d layers by a constant factor. This leaves us with two hyperparameters: While the dependence of the results on n_{out} is discussed in the results section and Figures 5 and 6 of the manuscript, we have chosen d by hyperparameter optimization. This procedure is described in the revised methods section.

- *A more specific example is the peptide neural network that had 30-10-3-3-2-5 layers, including a bottleneck of two nodes before the output layer that correspond to torsion angles. I would like the authors to explore what happens in terms of accuracy when the neural network design changes when it is not guided by expert knowledge (do not introduce a bottleneck layer based on no. of torsions). This extra information will also demonstrate how sensitive the network performance is to minor nuances to network architecture.*

Indeed this network is a specific choice with the sole aim to have a bottleneck with two neurons, whose activations can be correlated with the ϕ and ψ backbone torsions that are known to be important coordinates in this system. In general, the network architecture was chosen by selecting the number of output nodes, n_{out} , and the network depth, d , using hyperparameter optimization of the VAMP-2 score (see also next answer). For the alanine dipeptide VAMPnets we have reported the dependence of the score on n_{out} in Fig. 5d.

We have also added the new Supplementary Fig. 2b reporting the training success rate (see results section for definition) as a function of the network depth, and discussed this result in the revised results section. It is seen that networks with 4-7 layers and with 6 or more output states perform well. Thus, the networks can be trained with a high success rate for a range of architectures. Note that the network training is quite fast and we repeat 100 training repetitions for each result (see below), such that even success rates of 10-20% are acceptable.

- *How do you determine the number of states in the final softmax layer? It seems to be the authors are setting the final layer based on what they are "expecting" to observe. How does the model accuracy fare when that number of final states is set to an "incorrect" number?*

As shown in Fig. 5d, the VAMP score depends on the number of output states, but it reaches a plateau when all the states that are metastable at the chosen lag time are resolved. This is an effective automatic selection criterion. However, the user can choose to have a different number of output states in order to enforce a certain level of granularity - see Figures 5a-c and Figure 6a. We have revised the description in the caption of Fig. 5.

If the number of states is too small the discrepancy between prediction and estimation observed in the Chapman-Kolmogorov test will increase. If the number of states is too large the network will

split the fundamental states into several smaller states, which makes the resulting Markov model more difficult to interpret.

- *The network has 2 lobes, are they trained independently from one another (i.e. do the weights of lobe 1 share the same values as the weights of lobe 2)?*

In our examples, the two lobes have equal weight, i.e. they are just clones of the same network. But VAMPnets are not restricted to this choice. We added a clarification at the beginning of the revised results section.

- *Why was dropout only used for the first 2 layers? The usual practice is to apply dropout to all layers.*

We tested different values of dropout (between 0 and 50%) for all the layers in the network, but found the setup described in the paper to have the best performance. We assume this result to be caused by the “cone” shape of our network lobes, where most nodes are in the first two layers (and thus likely much redundancy in the weights), whereas rather few nodes are in the layers close to the output. It is reasonable that the layers close to the output do not use dropout, as this would effectively reduce the degrees of freedom required to represent the Koopman singular functions represented by the network output.

- *Can the trained neural network for NTL9 protein folding be directly used to generate MSMs of other proteins simulated under the same settings? Specifically, can the NTL9 neural network be used in inferencing mode (without retraining) on the other dozen or so proteins that DE Shaw simulated? If the results are not good, the authors should look into transfer learning using the NTL9 network.*

The current implementation of the neural network is strongly problem-related; specifically, it requires an unchanged order of the input coordinates and a constant input size, hence the feature transformations learned in the present paper are not transferable to other molecules.

However, this idea is however very interesting and important. We have thus discussed it in the revised conclusions as a starting point for future studies.

- *The authors mentioned in their abstract that their method "are competitive or outperform state-of-the-art Markov modeling methods". While they have provided convincing qualitative evidence of how VAMPnets can be used in place of MSMs methods, it is unclear how it is outperforming current MSM methods. The authors should perform a more quantitative comparison based on some error metric, or they should remove any claims of their methods performance relative to MSMs.*

Figure 6b and c already showed that for NTL9, a five-state VAMPnet performs equally well to a 40-state MSM – in this case the VAMPnet can be considered superior, because it achieves the same accuracy with a smaller and thus easier-interpretable model.

In the revised version we have also added a systematic comparison of VAMP-2 validation scores between VAMPnets and MSMs for alanine dipeptide (Revised Fig. 5d). It is seen that the MSM VAMP-2 scores are significantly worse than by VAMPnets when less than 20 states are employed. Clearly, MSMs will succeed when sufficiently many states are used, but in order to obtain an interpretable model those states must again be coarse-grained onto a fewer-state model, while VAMPnets directly produce an accurate model with few-states.

- *I would like the authors to address the time it takes to train the model (preferably the time including model exploration and network design as well). This should be stated in comparison with the current manual engineering process of creating MSMs.*

The training time depends on the problem and ranges between 20 and 180 seconds on a single NVIDIA GeForce GTX 1080 GPU for the systems studied in this paper. We have specified these results in the revised Methods section.

As we ran many training repetitions to compute average validation scores for each hyperparameter setting, this results in a few GPU days for hyperparameter selection per system. However, this effort can be trivially parallelized on a GPU cluster.

It is difficult to comparing these figures with the manual engineering time required to create an MSM. For the systems studied in the present paper, an MSM expert can obtain an MSM with comparable quality in 1-2 days of work. However, given the experiences we have made with a large user community in our PyEMMA workshops, this is not the typical scenario. Less experienced users often get stuck for very long time as a result of suboptimal choices made somewhere in the pipeline.

While recent efforts in MSM hyperparameter selection have certainly helped to reduce this frustration, our feeling is that VAMPnets make this procedure much more straightforward, as most of the MSM pipeline is integrated into the single step of neural network optimization.

- *Was there any k-fold cross-validation performed at all? I have the impression the authors did a 90/10 train/validation split only did not do k-fold cross-validation. This is problematic, the authors should be doing at least 5-fold cross validation to ensure their results are robust and not because they were "lucky" in splitting the data "correctly". Also, how was the split between training and validation performed? Did they do a time-index split, or did they randomly shuffle the frames and then do the split?*

Apologies for the incomplete explanation: For every choice made and every result reported we performed 100 optimization runs. In each run, the 90% / 10% split of data into training and validation sets was made randomly. The validation score is obtained as an average over these shuffled data splits. Similar as cross-validation, this approach ensures that the results are robust with respect to data splitting. We have described this procedure in the revised Methods section.

- *It seems that the authors performed some guided search on the hyper-parameters using the VAMP-2 validation score, however they now risk overfitting the hyper-parameters (aka neural network design) against the validation set, and I presume the validation set is what the authors are using to present all the results in the various figures. The authors should perform additional experiments but using a 3-way train/validation/test split. Keep the same protocol, but evaluate the final results on a separate test set that is not used in both the training of the model, or during hyperparameter optimization.*

This is a different aspect of the same concern as in the previous question, and it is thus also addressed by our shuffle-split approach (see previous answer)

REVIEWERS' COMMENTS:

Reviewer #1 (Remarks to the Author):

The revised manuscript addresses my concerns.

Reviewer #2 (Remarks to the Author):

The authors have addressed all my comments.

Reviewer #3 (Remarks to the Author):

The authors have satisfactorily addressed the concerns raised in my previous comments.